# Putative Novel Serotypes ‘33’ and ‘35’ in Clinically Healthy Small Ruminants in Mongolia Expand the Group of Atypical BTV

**DOI:** 10.3390/v13010042

**Published:** 2020-12-29

**Authors:** Christina Ries, Tumenjargal Sharav, Erdene-Ochir Tseren-Ochir, Martin Beer, Bernd Hoffmann

**Affiliations:** 1Friedrich-Loeffler-Institut, Federal Research Institute for Animal Health, Südufer 10, 17943 Greifswald-Insel Riems, Germany; christina.Ries@fli.de (C.R.); Martin.Beer@fli.de (M.B.); 2School of Veterinary Medicine, Mongolian University of Life Sciences, Khan-uul District, Zaisan 17024, Mongolia or tumee@muls.edu.mn (T.S.); erkavet@muls.edu.mn (E.-O.T.-O.)

**Keywords:** atypical BTV, putative novel serotypes, BTV in Mongolia, BTV in sheep and goats, BTV

## Abstract

Between 2015 and 2018, we identified the presence of three so-far-unknown Bluetongue virus (BTV) strains (BTV-MNG1/2018, BTV-MNG2/2016, and BTV-MNG3/2016) circulating in clinical healthy sheep and goats in Mongolia. Virus isolation from EDTA blood samples of BTV-MNG1/2018 and BTV-MNG3/2016 was successful on the mammalian cell line BSR using blood collected from surveillance. After experimental inoculation of goats with BTV-MNG2/2016 positive blood as inoculum, we observed viraemia in one goat and with the EDTA blood of the experimental inoculation, the propagation of BTV-MNG2/2016 in cell culture was successful on mammalian cell line BSR as well. However, virus isolation experiments for BTV-MNG2/2016 on KC cells were unsuccessful. Furthermore, we generated the complete coding sequence of all three novel Mongolian strains. For atypical BTV, serotyping via the traditional serum neutralization assay is not trivial. We therefore sorted the ‘putative novel atypical serotypes’ according to their segment-2 sequence identities and their time point of sampling. Hence, the BTV-MNG1/2018 isolate forms the ‘putative novel atypical serotype’ 33, the BTV-MNG3/2016 the ‘putative novel atypical serotype’ 35, whereas the BTV-MNG2/2016 strain belongs to the same putative novel atypical serotype ‘30’ as BTV-XJ1407 from China.

## 1. Introduction

The vector-transmitted Bluetongue virus (BTV) is the prototype virus of the genus *Orbivirus*, family *Reoviridae*, and can cause a systemic haemorrhagic fever, the bluetongue disease, especially in sheep [1,2,3]. The viral genome of the double-stranded RNA virus is comprised of 10 genome segments encoding for seven structural (VP1 to VP7) and five non-structural (NS1 to NS4, NS3A) proteins [4]. In the 1960s, the typing of bluetongue strains began in South Africa with the help of virus neutralisation assays [5]. Since then, the virus neutralization tests (VNT) have been recognised as the reference method for serotype identification and 24 classical BTV serotypes and three atypical BTV serotypes could be serologically defined [2,6,7,8]. The VP2 (Seg-2) as the most variant BTV protein is part of the outer capsid layer and was identified to play the major role in serum neutralization and therefore serotype specificity [2,9,10]. The identification of the serotype plays an essential part in disease control, as commercially available BTV vaccines on the market induce serotype-specific protection. In recent years, an increasing number of novel ‘serotypes’ have been described besides the discovery of BTV-25 in Switzerland [11], BTV-26 in symptomatic sheep in Kuwait [12], and BTV-27 in asymptomatic goats in Corsica [8]. Two putative novel serotypes—BTV-XJ1407 from China [13] and BTV-X ITL2015 from Italy [14]—were serologically characterized and described at the molecular level, however, they were not serotyped. Furthermore, to mention a few, BTV strains were isolated from a Sheep pox vaccine in Israel [15,16,17], from an Alpaca in South Africa [18], and the Tunisian BTV-Y TUN2017 [19]. All these novel atypical strains could not be adequately typed by serum neutralisation methods and consequently assignment to a specific serotype was not possible. Nevertheless, in 2011, Maan and colleagues [7] proposed cut-offs for nucleotide (nt) and amino acid (aa) identities in segment-2 of the BTV genome as an addition to the traditional typing methods [7,20]. The identity levels were defined firstly for strains belonging to different serotypes (maximum identity and minimum variation), and secondly for strains belonging to the same serotype (minimum identity and maximum variation). Firstly, strains of different serotypes can show up to 71.5% nt identity and 77.8% aa identity. Secondly, the described minimum levels of sequence identities in segment 2 and VP2 of strains belonging to the same serotype were defined with 68.4% in nt and 72.6% in aa. Looking at the major western and eastern topotypes separately, the maximum nucleotide variation within the same serotype in segment 2 decreases to 21.8% nt and to 13.9% in aa variation [7,20,21].

In China, BT disease/BTV is endemic with multiple serotypes circulating. Until 1997, only BTV-1, BTV-2, BTV-3, BTV-4, BTV-9, BTV-12, BTV-15, and BTV-16 were isolated from cattle [22]. In the subsequent years, leading to 2012, the presence of BTV-5, BTV-7 and BTV-24 in addition to the already known serotypes was revealed [22]. Analysis also showed that the Chinese BTV-7 strain carried not only segments from BTV strains of China, but also of African strains [23]. Likewise, for the BTV-21 strain isolated in 2018, genetic reassortment with BTV-16 was also proposed [24]. Furthermore, in 2014 the strain BTV-XJ1407 was isolated from clinical healthy sheep and goats in Xinjiang and was determined to be an atypical BTV due to the inability to be serotyped by the existing 1 to 24 reference sera [13]. Nevertheless, the sequence analysis suggested that BTV-XJ1407 is most likely a novel BTV serotype or belonging to BTV-25 or 27 with highest segment 2 sequence identities to BTV-27 (75% in nt/78% in aa) [13].

Nevertheless, the studies of BTV in China also included the province “Inner Mongolia”, a region that borders Mongolia directly to the south. Interestingly, the largest breeding population of sheep and goats in China was found to be in the province “Inner Mongolia” with mainly confined (extensive, mobile grazing systems) and extensive husbandry systems [25]. In this region, BTV antibodies are sero-prevalent in sheep, goats and cattle, a finding consistent with the existence of at least 36 species of blood-feeding midges [25]. For Mongolia itself, a landlocked republic bordering China and Russia, not much data are available about the circulation of different BTV serotypes. However, more than 60 million sheep and goats, and around 5 million cattle and yaks were kept in over 1.5 million km^2^ steppe, semi-desert and mountains by 2019 [26]. The presence of seropositive small ruminants and blood-sucking midges strongly suggested the occurrence of BTV also in Mongolia [27,28,29]. A serosurveillance study revealed that 86% of the goats were positive for BTV group-specific antibodies, followed by sheep with 51% and cattle with 9% [27,28].

In our study, we investigated the BTV situation in Mongolian livestock by analysing selected samples of goats and sheep in the Northern provinces Tov, Zavkhan, Selenge and Darkhan-Uul. We have successfully identified three novel atypical BTV strains in clinical healthy small ruminants. Propagation in cell culture was successful, and we used the isolates for further characterization. Whole genomes were generated and phylogenetically analysed. We proposed an order of the atypical BTVs as ‘putative novel serotypes’ for a more uniform nomenclature on the basis of genetics, especially since typing via VNT was not feasible. Consequently, two of the three newly discovered Mongolian strains of our study, BTV-MNG1/2018 and BTV-MNG3/2016, are classified as new BTV ‘putative novel serotypes’ 35 and 33, whereas BTV-MNG2/2016 is part of ‘putative novel serotypes’ 30.

## 2. Materials and Methods

### 2.1. Sampling in Mongolia

Blood and serum samples of 142 nomadically kept animals (34 sheep, 102 goats and 6 cattle) were collected in Tov, Darkhan-Uul, Selenge, and Zavkhan provinces of Mongolia and were shipped in two deliveries to the Friedrich-Loeffler-Institut (FLI, Isle of Riems, Germany) for further analyses. The samples were shipped in a polystyrene box with cooling racks, but not frozen. The first shipment in 2016 contained 26 EDTA blood samples, which were collected in 2015 and 2016 in Mongolia. The second shipment of 2018 contained in total the EDTA and serum samples of 116 animals, thereof 29 sheep and 87 goats, which were sampled in between December 2017 and March 2018. Sampling sites are shown in Figure 1.

### 2.2. Cell Culture Isolation of Virus In Vitro

From the first shipment, 14 blood samples (of 12 goats and 2 sheep) were selected for virus isolation, whereas from the second shipment, 27 blood samples (26 goats and 1 sheep) were chosen. All blood samples from naturally and experimentally infected goats were processed identically for the successful virus isolation experiments on BSR cells. Five hundred microliters of EDTA blood was centrifuged (6797 g) for 2 min and the red blood cells were washed twice in 1 mL PBS and finally diluted in 500 µL PBS prior to lysis by 20 s ultrasound treatment with a frequency of 20 kHz (Sonifier 450, Branson Ultrasonics, CT, USA). Unwashed blood of the experimentally infected goat was sonicated directly before virus isolation experiments. BHK-21 (BSR/5) cells (FLI cell culture collection number RIE0194) in T25 cm^2^ cell flasks were incubated initially for 3 h at 37 °C using the Eagles cultivation medium with Earles and Hank’s salts with non-essential amino acids (FLI intern medium number ZB5d) supplemented with 10% FCS (fetal calf serum). BHK-21 (BSR/5) cells were first described in the context of the virus isolation of BTV-27 [8]. Afterwards, the cells were inoculated with 200 µL of the processed blood preparations for 2 h. The BTV-MNG1/2018 field EDTA blood sample delivered a Cq-value of 27.4 in the Pan-BTV-S10 RT-qPCR, whereas the BTV-MNG2/2016 EDTA blood derived from the experimental infection showed the Cq-value of 23.1 and the BTV-MNG3/2016 field EDTA blood of 25.3 Cq-values. After the incubation period, the blood inoculum was removed and flasks replaced with medium supplemented with 10% *v*/*v* FCS and antibiotics in double standard concentration (20,000 µg/mL Penicillin, 20,000 units/mL Streptomycin, 10 mg/mL Gentamicin, 250 µg/mL Amphotericin B (Thermo Fisher Scientific, Waltham, MA, USA). After 3 to 5 days of incubation at 37 °C, the supernatant of the infected cell flask was retained, whereas the BSR cell monolayer was detached from the flask by using 1 mL of trypsin. The trypsinated cells were mixed with 5 mL of the retained supernatant of the respective cell flask. In a next step, 3 mL of the cell-trypsin-supernatant suspension were transferred to a new T75 cm^2^ cell flask with fresh BSR cells grown for 3 h. With this described method, three passages on BSR cells were performed and the success of virus replication was confirmed by genomic load estimated by RT-qPCR. The virus isolation procedure is shown in Figure 2.

Moreover, the virus presence was confirmed by the positive signal in the immune fluorescence test. Therefore, BSR cells were incubated for 4 h in 96-well cell culture plates and infected with the BTV-MNG1/2018, BTV-MNG2/2016 and BTV-MNG3/2016 virus suspension. After 4 days of incubation at 37 °C and 5% CO2, CpE was visible and infected and non-infected BSR cells were fixated with 100 μL ice-cold Acetone-Methanol 1:1 for 10 min. BSR cells were blocked with 100 μL ROTI^®^Block solution (Roth Chemie GmbH, Karlsruhe, Germany) for 30 min to reduce non-specific reaction. Then, we added 100 µL of the respective BTV-Mongolian rabbit immune serum diluted with ROTI^®^Block solution in the ratio 1:200 (1-h incubation). For the secondary antibody reaction, Goat anti-Rabbit IgG (Alexa Fluor^®^ 488, Abcam, UK) was prepared at a dilution of 1:1000 in ROTI^®^Block solution and 100 μL were added to each well (1-h incubation). Fluorescence signalling was analyzed using an Axio Vert.A1 microscope (Zeiss, Oberkochen, Germany) with an HXP 120 V fluorescent light source.

### 2.3. Propagation In Vitro

Virus propagation for the BTV-MNG1/2018 strain was tested on the mammalian cell lines BHK-21 (CT (clone Tübingen)) cells (FLI cell culture collection number RIE0164) and MDBK cells (FLI cell culture collection number RIE0261), and furthermore on the insect-derived KC-cell line (FLI cell culture collection number RIE1062).

### 2.4. Serological Profile

#### 2.4.1. Production of Antisera in Rabbits

Six New Zealand white rabbits, two for each strain of BTV-MNG1/2018, BTV-MNG2/2016 and BTV-MNG3/2016, were immunised with binary ethyleneimine (BEI)-inactivated full-virus BSR cell culture material in the insect-free high containment facilities of the FLI. Before inactivation, the virus preparations had titres of 10^5.7^ CCID_50_/mL (Cell culture infectious dose) for BTV-MNG1/2018, 10^3.6^ CCID_50_/mL for BTV-MNG2/2016 and 10^4^ CCID_50_/mL for BTV-MNG3/2016. BEI inactivation based on a standard procedure [31] and the antigen preparations were stored at −80 °C until use. The success of the inactivation procedure was confirmed by three serial passages of a virus isolation procedure and RT-qPCR. The rabbits were immunized with three applications of 1 mL inactivated antigen mixed with 100 µL of Polygen adjuvant (MVP Adjuvants^®^, Omaha, AL, USA) at intervals of 2 or 3 weeks and the serum was tested in antibody cELISA on day 0, 14, 28, and 42 dpi during the animal trial. The final serum was collected at 56 dpv (days post vaccination). The respective experimental protocols were reviewed by the state ethics commission and approved by the competent authority (State Office for Agriculture, Food Safety and Fisheries of Mecklenburg-Vorpommern, Rostock, Germany; Ref. No. LALLF M-V/TSD/7221.3-2-042/17).

#### 2.4.2. ELISA

All serum samples of field and experimental specimens were screened for group specific antibodies using a commercial cELISA targeting the VP7 (ID Screen^®^ Bluetongue Competition, ID-Vet, France) according to the manufacturer’s instructions [32]. Samples with ≤50% of negativity compared to the negative control (S/N) were considered as positive, samples with ≥50% S/N as negative.

#### 2.4.3. Virus Neutralisation Test

A virus neutralization test (VNT) was performed for the detection of serotype-specific neutralizing antibodies. The used BTV-MNG1/2018 strain was passaged 4 times on BSR cells, 7 times on Vero cells and another 14 times on BSR cells. The BTV-MNG2/2016 and BTV-MNG3/2016 strains were used after three passages on BSR cells, 1 passage on Vero cells and 14 passages on BSR cells. In the last passage, the content of a T1700 cm^2^ cell roller flask was pelleted via centrifugation (2862× *g*) and re-suspended in 60 mL medium. For the VNT stock viruses of BTV-MNG1/2018, BTV-MNG2/2016 and BTV-MNG3/2016, titres are 10^5.3^, 10^5.0^ and 10^5.5^ CCID_50_/mL and can be defined, respectively. VNTs for all three strains were performed with the reference sera of classical BTV serotypes 1 to 24 (generated in guinea pigs or rabbits) and were sera reactive against BTV-25 [33], BTV-26 [7], BTV-27x [8], and BTV-28 [16] as well as against the three Mongolian strains. Plates were incubated for 1 h at 37 °C before overnight incubation at 4 °C. The following day, 100 µL of a BSR cell suspension of approximately 30,000 cells/100 µL was added per well. After incubation for 3–5 days at 37 °C, all wells were scored for a cytopathic effect (CpE). The neutralization titre was determined as the dilution of serum giving 100% neutralization and calculated using the Spearman and Kärber method [34,35].

### 2.5. Experimental Infection of Goats

The experimental inoculation with BTV-MNG2/2016 was conducted for receiving fresh EDTA blood for the isolation of BTV-MNG2/2016 in cell culture, because virus isolation remained unsuccessfully with the field blood samples. We chose goats for the experimental inoculation, because two of the three field EDTA blood samples for inoculation were goat samples, whereas one blood sample was of a sheep. Three male, 6-months-old Thuringian goats, seronegative for BTV, were kept in the insect-free high containment facility of the FLI for experimental inoculations. The three goats were inoculated subcutaneously at two different injection sites with 3 × 700 µL of EDTA-blood samples positive for BTV-MNG2/2016 from three different hosts (each goat received in total 2.1 mL). The EDTA blood samples used for experimental inoculation delivered quantification cycle (Cq) values of 26.5 to 32.7 in the Pan-BTV-S10 RT-qPCR. EDTA blood samples with a reactivity in the BTV antibody cELISA were washed with PBS two times to remove neutralizing antibodies before inoculation. All blood preparations were injected subcutaneously on two different injection sites. Furthermore, the goats were monitored daily for clinical signs (the applied clinical score is added to the Appendix A) and EDTA blood and serum were taken regularly on days 0, 3, 5, 7, 10, 11, 12, 13, 14, 17, 19, 21, 24, 28, and 31 post infection. The respective experimental protocols were reviewed by the state ethics commission and approved by the competent authority (State Office for Agriculture, Food Safety and Fisheries of Mecklenburg-Vorpommern, Rostock, Germany; Ref. No. LALLF M-V/TSD/7221.3-2-010/18).

### 2.6. RNA Extraction and RT-qPCR

Viral RNA of all EDTA blood samples was extracted either manually using the QIAamp Viral RNA Mini kit (Qiagen, Hilden, Germany) or the NucleoMagVET kit (Macherey-Nagel, Düren, Germany) on the KingFisher platform (King-Fisher Flex magnetic particle processor, Thermo Fisher Scientific, Waltham, MA, USA). The RNA was amplified using the Pan-BTV-S10-RT-qPCR recommended of the OIE [36] for samples of both shipments and the Pan-S5-RT-qPCR for samples of shipment 1 [37]. For further screenings, specific RT-qPCRs for BTV-MNG1/2018, BTV-MNG2/2016 and BTV-MNG3/2016 were developed (shown in Appendix A). Results were considered positive when Cq-values were <40. RT-qPCR reactions with blood samples of the first shipment were run with the qScript XLT One-Step RT-qPCR ToughMix of QuantaBio (Beverly, USA). The composition of the RT-qPCR reactions was 2.75 μL of RNase-free water, 6.25 μL of qScript XLT One-Step RT-qPCR ToughMix, 1 μL of primer-probe-mix-FAM (7.5 pmol of each primer and 2.5 pmol of the FAM probe) and 2.5 µL template RNA with the temperature profile of 10 min at 50 °C, 1 min at 95 °C, followed by 45 cycles of 10 s 95 °C, 30 s 57 °C and 30 s 68 °C. For the second shipment and all further analysis, the RT-qPCR reactions were run with the AgPath-ID™ One-Step RT-PCR Reagents of Applied Biosystems™ (Waltham, MA, USA) with the integration of an heterologous control system as process control using an internal control system [38]. The composition of these RT-qPCR reactions was 1.25 μL of RNase-free water, 6.25 μL of 2× RT-PCR buffer, 0.5 μL of RT-PCR enzyme mix, 1 μL of primer-probe-mix-FAM, 1 μL of EGFP-mix1-HEX (2.5 pmol of each primer and 2.5 pmol of the HEX-probe) and 2.5 µL template RNA. The temperature profile used was 10 min at 45 °C, 10 min at 95 °C followed by 42 cycles of 15 s at 95 °C, 20 s at 56 °C and 30 s at 72 °C. The 2.5 µL of template RNA was heat denatured for 3–5 min at 95 °C and immediately cooled down in liquid nitrogen. All RT-PCRs were run on the CFX 96 real-time PCR cycler (Bio-Rad, Hercules, CA, USA) and fluorescence values (FAM, HEX) were collected during the annealing step. The specificity of each single Mongolian assay was evaluated in silico by BLAST search (https://blast.ncbi.nlm.nih.gov) and in vitro using available viral RNAs of all 24 classical BTV serotypes and further atypical BTV serotypes (BTV-26, three variants of BTV-27 and BTV-28).

### 2.7. Sequence Analysis

The sequences of the three Mongolian BTV strains propagated in cell culture were generated using the HTS-SISPA technology like previously described for BTV-25-GER2018 [33,39]. The amplified and purified ds cDNA was sent to Eurofins Genomics (Ebersberg, Germany) for sequencing on an Illumina platform. Raw data as fastq files were trimmed and assembled by mapping to the BTV-XJ1407 reference sequences with the following accession numbers: KR815991 (Seg-1), KR061882 (Seg-2), KR085413 (Seg-3), KR085414 (Seg-4), KR085415 (Seg-5), KR061883 (Seg-6), KR085416 (Seg-7), KR085417 (Seg-8), KR085418 (Seg-9), and KR085419 (Seg-10) using the Geneious software v2019.2.3 (Biomatters Ltd., Auckland, New Zealand). For phylogenetic analyses, a multiple alignment of BTV-sequences was performed by using the MAFFT alignment feature in the Geneious software. We included the identical BTV-strain selection, as used in the publication of BTV-X-ITL2015 and BTV-25GER-2018 [14], and additional atypical BTV strains. Phylogenetic trees of each of the 10 segments of the three strains were created with MegaX using the genetic distinction model Tamura–Nei and tree-built method Maximum likelihood [40]. To assess the robustness of individual nodes on the phylogenetic trees, we performed a bootstrap analysis with 100 replications. Furthermore, the consensus sequences of each of the 10 segments of the three Mongolian strains were blasted against the nt/aa database on NCBI to identify the nearest molecular neighbours. Sequences obtained in this study were submitted to NCBI with the following accession numbers: BTV-MNG1/2018 (accession numbers LR877337 to LR877346), BTV-MNG2/2016 (accession numbers LR877347 to LR877356) and BTV-MNG3/2016 (accession numbers LR877358 to LR877367).

## 3. Results

### 3.1. Sampling in Mongolia

EDTA blood samples (*n* = 142) collected from different ruminants in Mongolia in the years 2015, 2016 and 2018 were analysed. Of both shipments, samples positive for BTV genomes in the Pan-BTV-S10 RT-qPCR were from 0 of 6 cattle, 12 of 34 sheep and 95 of 102 goats. Here, Cq values of 29 to 36 could be ascertained. Analyses with real-time RT-PCR assay focused on the classical BTV serotypes 1–24 and the partial VP2 sequencing of the Pan-BTV-S10 RT-qPCR positive samples revealed that no classical BTV serotype could be detected. In contrast, three different novel BTV-strains, so called BTV-MNG1/2018, BTV-MNG2/2016, and BTV-MNG3/2016, were identified in 7 sheep and 58 goats. That means that for 42 Pan-BTV-S10 RT-qPCR positive samples with a low viral genome load, the serotype could not be defined, neither by serotype-specific real-time RT-PCR systems nor by sequencing. In 45 of the 65 serotyped samples, only one BTV strain (MNG1, MNG2 or MNG3) could be ascertained. BTV-MNG1/2018 was only found in goats, whereas BTV-MNG2/2016 and BTV-MNG3/2016 were found in both clinical healthy sheep and goats (shown in Table 1). Mixed-strain detections were seen in 1 sheep and in 19 goats (shown in Table 2).

### 3.2. Virus Isolation In Vitro

BTV-MNG1/2018 and BTV-MNG3/2016 were successfully isolated on BSR cells from the original blood samples from the goats in Mongolia, whereas BTV-MNG2/2016 could not be isolated from the original blood samples from the goats and sheep in Mongolia. In detail, virus isolation succeeded for BTV-MNG1/2018 from one field-infected goat, for BTV-MNG2/216 from one experimental infected goat and for BTV-MNG3/2016 from two field infected goats. Hence, the Mongolian strains were isolated in cell culture from goat EDTA blood samples, but not from sheep. Fortunately, both washed and unwashed EDTA blood of the BTV-MNG2/2016 experimentally-infected goat led on BSR as well as on Vero cells to virus isolation.

After the third cell culture passage, CpE was observed for the BTV-MNG1/2018 and BTV-MNG3/2016 field EDTA blood sample, however, no CpE was observed for the BTV-MNG2/2016 field EDTA blood sample from Mongolia. Nevertheless, CpE was observed for the BTV-MNG2/2016 EDTA blood sample (washed and not washed) of the experimental infected goat. The successful propagation of the virus could be confirmed with decreasing Cq values in the RT-qPCR in subsequent passages, alhough not for the subsequent passages of the BTV-MNG2/2016 field EDTA blood samples, which is in line with the observation of no CpE.

### 3.3. Virus Propagation

Virus propagation of BTV-MNG1/2018 was successful on Vero- and BHK cells, but not on MDBK cells or KC-cells. For all three Mongolian strains, the pelleted infected cell culture material achieved higher titres in CCID_50_ (10^2^ CCID_50_ mean difference in titre) as well as in the Cq-value in RT-qPCR (7.1 mean difference in Cq-value) compared to the clarified cell culture supernatant.

### 3.4. Serological Profile

#### 3.4.1. Production of Antisera in Rabbits

Polyclonal antisera reactive against the Mongolian strains were generated from six immunized rabbits with increasing positivity in BTV antibody cELISA. On day 14 of the animal trial, five of the six rabbits were cELISA positive, whereas on days 28 and 42 all rabbits were strong cELISA positive. The rabbit sera on day 56 were positive in the cELISA for group-specific BTV antibodies with log2 cELISA titres for BTV-MNG1/2018 and BTV-MNG2/2016 of 1:4, and for BTV-MNG3/2016 of 1:8. Nevertheless, no neutralizing antibodies could be detected for the rabbit sera due to an incomplete neutralization pattern in the VNT.

#### 3.4.2. Virus Neutralization

The VNTs using BTV-MNG1/2018 and BTV-MNG3/2016 performed with positive sera of the rabbit immunisation trial on cELISA, revealed an incomplete neutralization at the 1:10 dilution step of the serum dilution and were negative at higher dilutions 1:20, 1:40, 1:80, 1:160, 1:320, 1:640, and 1:1280. The positive rabbit serum for BTV-MNG2/2016 on cELISA neutralized the BTV-MNG2/2016 virus strain in two of three wells at the 1:10 dilution step, whereas the serum of the experimental BTV-MNG2/2016-infected goat did not neutralise the three wells at the 1:10 dilution step. The reference sera BTV 1 to 24, as well as the sera of BTV-25-GER2018, BTV-26, BTV-27x, BTV-28, BTV-MNG1/2018, BTV-MNG2/2016, and BTV-MNG3/2016 did no neutralize the Mongolian strains. An exception was the BTV-MNG1/2018 anti-serum, which neutralized the BTV-MNG2/2016 virus strain in one of three wells at a dilution of 1:10.

### 3.5. Animal Experiments

Experimental inoculation with the BTV-MNG2/2016 field blood samples led to successful infection in one of the three animals (goat #18). Viral RNA detected started from 7 dpi (days post infection) and peaked on 14 dpi with a Cq value of 23.1. The sera of the positive goat from days 7, 14, 21, and 31 dpi showed a trend of seroconversion and was confirmed to be seropositive on 31 dpi. Both other experimentally-infected goats remained BTV genome and antibody negative throughout the study (Table 3). All three goats did not show any clinical symptoms or fever throughout the trial.

### 3.6. Genome Analysis

The sequences of all 10 segments of the virus strains BTV-MNG1/2018, BTV-MNG2/2016 and BTV-MNG3/2016 were established and phylogenetic analyses performed against BTV strains of representative serotypes (Figure 3). The BLASTn results of the nucleotide and amino acid sequences of the complete coding sequences of all 10 segments of the three Mongolian strains are shown in Table 4. Of all three Mongolian strains, the nearest neighbours for all segments were found to be representatives of atypical BTVs.

In detail, for segment 2 of BTV/MNG1-2018, the nt identities for the BTV strains used in the phylogenetic tree were varying from 40.9% (BTV-15) up to 59.2% (BTV-20) for the classical serotypes 1 to 24. In the case of BTV-MNG2/2016, the nt identities for the BTV strains were varying from 41.1% (BTV-15) to 60.0% (BTV-24), and for BTV-MNG3/2016 from 40.5% (BTV-15) to 60.1% (BTV-20) for the classical serotypes. The identities in segment 2 with the atypical serotypes were in comparison higher for all three Mongolian strains. The nt identity for the atypical serotypes ranged from 56.0% (BTV-28) to 64.7% (BTV-26) for BTV-MNG1/2018, from 58.4% with BTV-28 for up to 81.4% with BTV-XJ1407 for BTV-MNG2/2016, and from 56.5% with BTV-28 for up to 69.9% with BTV-XJ1407 and 71.9% with BTV-MNG2/2016 for BTV-MNG3/2016.

The phylogenetic tree of segment 2 shows BTV-MNG2/2016 in close proximity to the BTV-XJ1407 strain and BTV-MNG3/2018 more distant but clearly in between the other atypical BTV strains. However, BTV-MNG1/2016 is in the same cluster as BTV-26 and SPvvvv/02. The nearest neighbour in the BLAST search is in line with the findings of the phylogenetic trees, showing the nearest neighbour of BTV-MNG2/2018 and BTV-MNG3/2016 to be BTV-XJ1407, whereas for BTV-MNG1/2016, the nearest neighbour was found to be BTV-26 in both nt and aa. For segment 6, the BTV-MNG2/2016 strains clustered with BTV-XJ1407, which is in line with the BLAST result in both nt and aa. The BTV-MNG1/2018 and BTV-MNG3/2016 strains are more distant to other strains and gathering with BTV-26, SPvvvv/02, BTV-Y TUN2017, BTV-X ITL2015 and two variants of BTV-27 in the phylogenetic trees. The BLAST reflects the phylogenetic tree and revealed the nearest neighbour in nt and aa for BTV-MNG1/2018 to be SPvvvv/02, whereas for BTV-MNG3/2016, the nearest neighbour in nt was BTV-27v02 and in aa BTV-X ITL2015. The phylogenetic trees for segments 1, 3, 4, 5, 7, 8, and 9 revealed that all three Mongolian strains are close to each other and to the two Chinese strains BTV-XJ1407 and V/196/XJ/2014, which is in line with the BLAST research of finding BTV-XJ1407 and V/196/XJ/2014 as nearest neighbours in nt and aa. In segment 10, BTV-MNG3/2016 and BTV-MNG1/2018 showed proximity with each other and with V/196/XJ/2014, whereas BTV-MNG2/2016 is close to BTV-27 v02 and BTV-X ITL2015. Likewise, the nearest neighbour found in BLAST research for BTV-MNG1/2018 and BTV-MNG3/2016 was V/196/XJ/2014 in nt and aa. In contrast, for BTV-MNG2/2016 the nearest neighbour wasBTV-27v02 in nt and BTV-X ITL2015 in aa.

### 3.7. Novel Putative Serotype Classification

BTV strains can be typed traditionally as serotypes by VNT based on selected reference serotypes 1–24, but also via molecular analysis. We sorted all atypical BTV according to their levels of sequence homologies of segment 2 towards other BTV strains, and sorted the (putative novel) serotypes chronologically to their collection date of the first identified BTV strain of the respective serotype. We used the following published sequence identities levels: Strains of different serotypes can show up to 71.5% nt identity and 77.8% aa identity, with a minimum of 28.5% nt variation and 22.2% aa variation between serotypes. The described minimum levels of sequence identities in segment 2 and VP2 of strains belonging to the same serotype were defined with 68.4% in nt and 72.6% in aa and consequently with a maximum of 31.6% nt variation and 27.4% aa variation [20,21]. An overview of the currently known atypical BTV strains and their assignment to ‘putative novel atypical serotypes’ is shown in Table 5. We sorted our novel Mongolian strains as well. The nucleotide sequence of the segment-2 of BTV-MNG1/2018 shared 68.1% identity with BTV-26 at, and the closest neighbour at amino acid level was likewise BTV-26 with 67.0%. It could not be classified as BTV-26 based on this method. Consequently, based on the sequence data analysis, BTV-MNG1/2018 represents the putative novel atypical serotype 35.

The nucleotide sequence and amino acid of the segment-2 of BTV-MNG2/2016 shared 85.0% identity with BTV-XJ1407 and at 85.8%. According to this high similarity on genome basis, BTV-MNG2/2016 is part of the same putative novel atypical serotype 30, like the Chinese BTV-XJ1407.

The nucleotide sequence of segment-2 of BTV-MNG3/2016 matches to 72.8% with BTV-XJ1407. The closest neighbour on amino acid level is likewise BTV-XJ1407 with 75.2%. The nucleotide variation of 27.2% between the nearest neighbour is close to the observed 28.5% minimum variation in nt described by Maan, et al. The second nearest neighbour of BTV-MMNG3/2016 was found to be BTV-Z ITL2017 in nt with 72.4% identity and BTV-27v01 with 74.4% in aa. Hence, the second nearest neighbour BTV-27v01 is nearly as close as the nearest neighbour BTV-XJ1407, but they are of different serotypes. This is also reflected in the phylogenetic tree, where the BTV-MNG3/2016 strain is on a distinct separate branch, not together with BTV-XJ1407 and BTV-MNG2/2016. Additionally, the sera of BTV-MNG3/2016 is not neutralizing BTV-MNG2/2016 (the same novel putative serotype as BTV-XJ1407), and the specific BTV-MNG3/2016 assays do not detect the other Mongolian strains. Therefore, we conclude that BTV-MNG3/2016 represents a putative novel atypical serotype 33. A similar decision was made for BTV-27v01, which showed 73.0% identity in nt with BTV-25, which is bigger than the suggested 68.4% of Maan et al. [7]. BTV-27v01 was declared as a novel serotype in agreement with the BTV community [8]. The calculations of Maan et al., [7] are good guidelines for classification of the novel atypical BTV strains, but are not to be understood as exact values, as they are based on calculations of sequence identities at the respective time point of research. The inter-variant identities between strains are significantly higher than 80% identity like for the three variants of BTV-25 and the three variants of BTV-27.

## 4. Discussion

Here we present the discovery of three novel BTV strains in Mongolia, BTV-MNG1/2018, BTV-MNG2/2016 and BTV-MNG3/2016. The three Mongolian strains were found to circulate in clinically healthy sheep and goats, which is characteristic of atypical BTV in general [14,36,43]. We could efficiently propagate BTV-MNG1/2018, BTV-MNG2/2016 and BTV-MNG3/2016 in cell culture and further characterize them. Molecular analyses showed the highest identities on both the nucleotide and amino acid sequence level with the group of atypical BTV strains, which are clearly distinct from the canonical “classical” serotypes 1–24. Furthermore, we suggested putative novel serotypes and sorted the atypical BTV strains according to their collection date of the sample and their sequence homologies in segment 2 to the other BTV strains. We grouped the, to our knowledge, currently 18 atypical strains in 11 different (putative novel) serotypes (25 to 35). BTV-MNG1/2018 and BTV-MNG3/2016 are representing new BTV putative novel serotypes 35 and 33. In contrast, due to its high similarities on the genome level, BTV-MNG2/2016 belongs to the putative novel serotype 30, together with the atypical Chinese BTV-XJ1407 strain.

Conversely, only a few studies are currently available regarding the BTV situation in Mongolia. In one report of the FAO from 2010 [30], the sero-prevalence of group-specific antibodies formed towards BTV infection in the Mongolian livestock was analysed via cELISA. Bluetongue virus antibodies could be detected in all Mongolian regions, so called ‘aimags’. The highest sero-prevalence was detected in goats with up to 86%, whereas in sheep (51%) and cattle (9%), the sero-prevalence was lower of the analysed Mongolian animals [27,28]. Hence, it is not surprising that we found also in our study a certain amount of BTV positive sheep and goats. Interestingly, we could not identify any classical BTV strain, but three novel atypical BTV strains.

So far, all atypical BTV strains were detected in goats and sheep only [7,8,11,13,14,33,42,44]. Similar to BTV-25, BTV-26, BTV-27, BTV-X ITL2015, and the Chinese BTV-XJ1407, the Mongolian strains did not cause clinical signs in the experimental condition [6,13,14,43,44]. BTV-MNG1/2018 was only found in goats, whereas BTV-MNG2/2016 and BTV-MNG3/2016 were found in both clinically healthy sheep and goats. Nevertheless, the small sample size from sheep could impose bias for the association with infection in sheep BTV-MNG1/2018-infected sheep. However, experimental BTV-27 inoculation did not cause a productive infection in sheep, and cattle inoculated with BTV-27v02 seem not to be susceptible [44]. On the other hand, BTV-26 antibodies were found to be circulating in cattle and dromedaries in Mauritania, suggesting susceptibility of cattle [45]. Hence, further pathogenesis studies of the three Mongolian strains themselves are necessary to rule out their potential susceptibility to other ruminant species like BTV-MNG1/2018 in sheep or all three strains in cattle.

Interestingly, 1 sheep and 17 goats were tested positive for more than one Mongolian BTV strain. In two goats, the RNA of even all three Mongolian strains were detected with the specific RT-qPCRs. These findings suggest that no cross-neutralising antibodies towards the Mongolian strains are produced, which is supported by the fact only an incomplete neutralization in the VNT was observed. This might indicate a lack of sufficient neutralizing antibodies, which was suggested for BTV-25-GER2018 as well [33]. Further studies are necessary to analyse the immune response of the host-caused BTV-MNG1/2018, BTV-MNG2/2016 and BTV-MNG3/2016, which might be poorly immunogenic. The transmission of BTV to the mammalian host is vector dependant [2]. Consequently, the propagation on mammalian cells as well as on a *Culicoides*-derived KC cells was possible for the classical BTV serotypes 1–24 [46]. Nevertheless, especially for the atypical BTV, alternative transmission ways have been discussed in the past since BTV-26 was found to be transmitted horizontally by direct contact [47] and BTV-26 and BTV-27 were not replicating in the KC cell line [44,47]. For other atypical serotypes like for TOV, BTV-Z ITL2017 and BTV-X ITL2015, virus isolation attempts were not successful [14,48]. Interestingly, a study group could identify small changes in NS3/NS3a or in the outer shell protein VP2 to influence the BTV vector competence shown for BTV-11 and BTV-26 [46].

In our study, the BTV-MNG1/2018 strain could be propagated on mammalian cell lines, but not on the insect-derived KC-cell line. It needs therefore to be further analysed if alternative transmission routes play a major role for the three Mongolian strains and the atypical BTV in general. Nevertheless, the comparative testing of infected cell pellet and cell supernatant suggests a strong cell association of all three Mongolian strains, as observed likewise for the classical BTV strains. In a further study, virus isolation of the three Mongolian strains on different cell types and with different approaches should be investigated.

The raising number of recently discovered atypical BTVs combined with the difficulties concerning their serotype assignment via VNTs requires a uniform nomenclature. Former studies on the comparisons of the segment-2 sequences revealed a complementary relationship between the respective BTV serotype and the Seg-2/VP2 sequence [49]. That enabled the classification of BTV ‘serotypes’ via sequence analysis of the Seg-2 [50]. In context with the publication of the 26 th BTV serotype, specific levels of sequence identities for isolates belonging to the same or different serotypes were defined. So, BTV serotypes belonging to the same serotype share an identity in Seg-2/VP2 of >68.4%/72.6% nt/aa. Between different serotypes identities of 40.5%/22.1% to 71.5%/77.8% nt/aa for the Seg-2/VP2 can be ascertained [7]. However, the term ‘serotype’ implies the capability of the serum to neutralise the respective virus and need to be tested in a virus neutralisation assay. Virus serotyping is furthermore very important for BTV control, as current BTV vaccines are all serotype-specific. In contrast, ‘Putative novel serotypes’ categorises BTV strains according to their segment 2 sequence, a method which is especially useful in the case of strains that could not be serotyped. Serotyping for the novel atypical BTV was in several cases not possible due to missing virus isolates and the according strain specific antisera (e.g., BTV-Z ITL2017, BTV-X ITL2015, and TOV). In addition, the observed incomplete neutralization of some viruses as described for BTV-25 GER2018 [33] and the presented Mongolian strains here reduce the expressiveness of performed serum neutralisation tests. Consequently, the BTV isolates BTV-28/1537/14 [15] and SPvvvv/03 [17] isolated from the sheep-pox vaccine are belonging to the ‘putative novel serotype’ 28, followed by the SPvvvv/02 isolate [17], likewise from a sheep-pox vaccine as putative novel serotypes 29. The Chinese BTV-XJ1407 forms the ‘putative novel serotype’ 30 [13] together with BTV-MNG2/2016, whereas the Chinese V196/XJ/2014 forms the putative novel serotype 31, followed by BTV-X ITL2015 and BTV-X ITL2015 strain 33,531 as ‘putative novel serotypes’ 32 [14]. BTV-MNG3/2016 forms the putative novel serotype 33; BTV-Y TUN2017, the ‘putative novel serotype’ 34; [19] and BTV-MNG1/2018 strain is classified as the ‘putative novel serotype’ 35. We did not include the BTV strains found in Alpacas in South Africa published in 2014 in a doctoral thesis [18] as no sequence information is currently available, and consequently, no genotyping could be performed.

The phylogenetic analysis revealed that the segments 1, 3, 4, 5, 7, 8, and 9 of all three Mongolian strains are closely related to BTV-XJ1407 and V/196/XJ/2014. This is not surprising as BTV-XJ1407 and V/196/XJ/2014 are both detected in China, which is a neighbouring country of Mongolia [13]. However, in segments 2, 6 and 10, some of the three Mongolian strains did reveal the nearest neighbour’s amongst BTV strains detected in more distant countries like Israel (SPvvvv/02 [17]), Kuwait (BTV-26 [7]), France (BTV-27v02 [42]), and Italy (BTV-X ITL2015). The global spread of BTV is a complex, multifactorial development driven by environmental and anthropogenic factors [51]. Furthermore, reassortment events can happen and are not limited to phylogenetically-related viruses [52]. This might have occurred for the mentioned segments 2, 6 and 10 of the mentioned Mongolian strains.

In summary, we identified three atypical BTV strains, BTV-MNG1/2018, BTV-MNG2/2016 and BTV-MNG3/2016 circulating in healthy sheep and goats in Mongolia. Virus isolation on mammalian cell lines was successful, and the complete coding sequences of each of the 10 segments of the three Mongolian strains was generated. Furthermore, we grouped the 18 atypical BTV strains, to our knowledge currently published with according sequence data, in 11 (putative novel) serotypes, based on the genome sequence identities of segment-2. Hence, we proposed the viruses belonging to ‘putative novel serotypes’. BTV-MNG1/2018 forms the putative novel serotypes 35; BTV-MNG2/2016 belongs to the putative novel serotype 30; and BTV-MNG3/2016 forms the putative novel serotype 33.

## Figures and Tables

**Figure 1 viruses-13-00042-f001:**
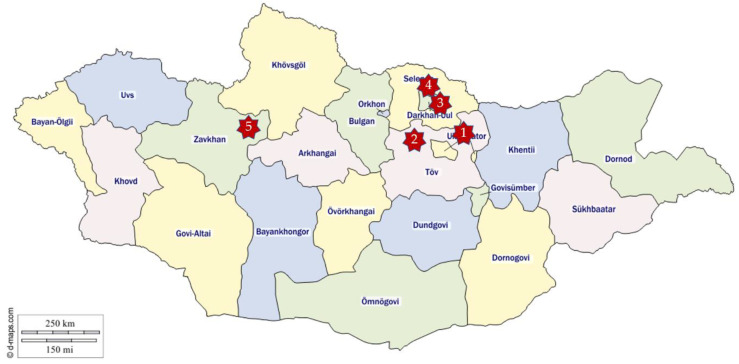
In this map of Mongolia, the sampling sites are indicated with red stars. Sampling site 1 was in the aimag TOV, sum Erdene. Sampling site 2 in the aimag TOV as well, but in the sum Ugtaal. The sampling site 3 was in the aimag Darkhan-Uul, sum Shariin Gol; sampling site 4 in the aimag Selenge, sum Javkhlant; and sampling site 5 was in the aimag Zavkhan, sum Tosontsengel. The map was taken from [30].

**Figure 2 viruses-13-00042-f002:**
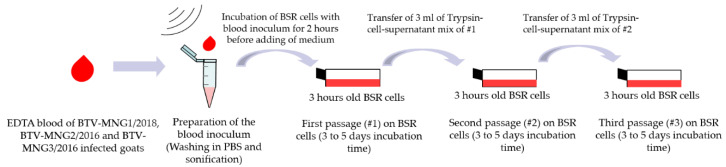
Shown is the virus isolation procedure of the three Mongolian strains on BSR cells.

**Figure 3 viruses-13-00042-f003:**
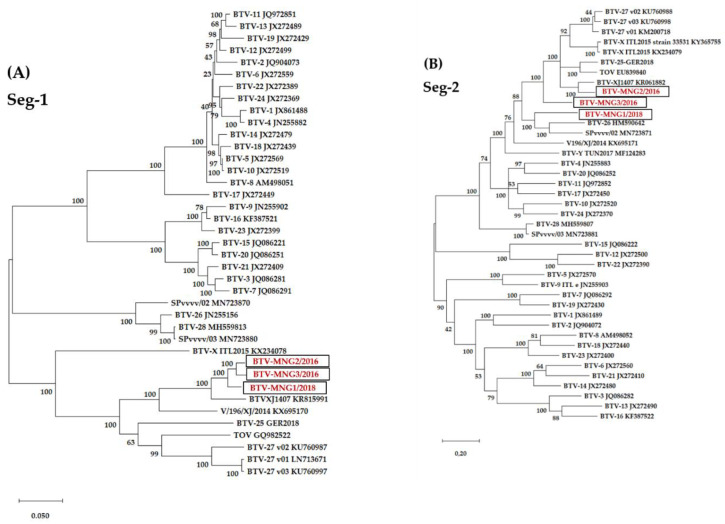
Phylogenetic analyses of the genome of the three Mongolian strains. The phylogenetic trees of each of the 10 segments were created with MegaX using the genetic distinction model Tamura–Nei and tree-built method Maximum likelihood including BTV strains representing the published BTV serotypes. We performed a bootstrap analysis with 100 replications. The black boxes surround the BTV-MNG1/2018, BTV-MNG2/2016 and BTV-MNG3/2016 sequences highlighted in red. Subfigure (**A**) shows Seg-1, (**B**) Seg-2, (**C**) Seg-3, (**D**) Seg-4, (**E**) Seg-5, (**F**) Seg-6, (**G**) Seg-7, (**H**) Seg-8, (**I**) Seg-9 and (**J**) Seg-10.

**Table 1 viruses-13-00042-t001:** Shown is the amount of positive sheep and goats in the different RT-qPCR assays of the EDTA blood divided in the two shipments from Mongolia to the FLI. The total number of positive sheep and goats of both shipments is given, as well as the percentage of positive sheep and goats of the in total 34 sampled sheep and 102 sampled goats. The number of sheep or goats positive in cELISA is given in brackets, whereas the ‘na’ indicates ‘not analyzed’ as serum samples were not available in the first shipment. The quality and volume of the serum samples did not allow further analysis in the VNT. Hence, no conclusion about serotype-specific antibodies can be made.

	Positive in (Thereof cELISA Positive)		Single BTV Genome Detection of (Thereof cELISA Positive)					
	Pan-BTV-S10 RT q-PCR		BTV-MNG1/2018		BTV-MNG2/2016		BTV-MNG3/2016	
Animal Species	Sheep	Goat	Sheep	Goat	Sheep	Goat	Sheep	Goat
First shipment	5 (na)	14 (na)	0 (na)	4 (na)	5 (na)	3 (na)	0 (na)	1 (na)
Second shipment	7 (0)	81 (23)	0 (0)	5 (1)	0 (0)	7 (1)	1 (0)	19 (5)
Total number	12	95	0	9	5	10	1	20
Percentage	35.3%	93.1%	0%	8.8%	14.7%	9.8%	2.9%	19.6%

**Table 2 viruses-13-00042-t002:** Shown is the number of sheep and goats with a positive BTV genome detection in more than one of the three different Mongolian strains. The results are divided in the two shipments from Mongolia to the FLI, but also the total number of positive sheep and goats summarized of both shipments is given. In 1 sheep and 19 goats, mixed-strain detections were seen.

	BTV Genome Detection of							
	BTV-MNG1/2018 & BTV-MNG2/2016		BTV-MNG1/2018 & BTV-MNG3/2016		BTV-MNG2/2016 & BTV-MNG3/2016		BTV-MNG1/2018, BTV-MNG2/2016 & BTV-MNG3/2016	
Animal Species	Sheep	Goat	Sheep	Goat	Sheep	Goat	Sheep	Goat
First shipment	0	3	0	2	0	0	0	1
Second shipment	0	4	0	2	1	6	0	1
Total number	0	7	0	4	1	6	0	2

**Table 3 viruses-13-00042-t003:** Overview of the RT-qPCR (Cq-value) and cELISA (S/N %) results after diagnostic inoculation of three goats with BTV-MNG2/2016 (#16, #17 and #18).

Goat	dpi	0 dpi	5 dpi	7 dpi	10 dpi	14 dpi	17 dpi	21 dpi	24 dpi	31 dpi
#16	RT-qPCR cELISA	no Cq	no Cq	no Cq 101	no Cq	no Cq 93	no Cq	no Cq 94	no Cq	no Cq 97
#17	RT-qPCR cELISA	no Cq	no Cq	no Cq 111	no Cq	no Cq 101	no Cq	no Cq 101	no Cq	no Cq 103
#18	RT-qPCR cELISA	no Cq	no Cq	37.6 105	29.6	23.1 87	24.0	25.4 52	25.9	25.9 43

**Table 4 viruses-13-00042-t004:** Nearest neighbours of BTV-MNG1/2018 (CDS (Coding sequence)), BTV-MNG2/2016, and BTV-MNG3/2016 determined with BLASTn and BLASTp analyses. The (nt/aa) indicates the nearest neighbours in nucleotide sequence (nt) and in amino acid (aa).

	BLAST Best Hits for BTV-MNG1/2018				
Segment/Protein (Accession No.)	Serotype (nt/aa)	Strain (nt/aa)	Accession No. (nt/aa)	Identity Level % (nt/aa)	Query Cover % (nt/aa) *
1/VP1 (LR877337)	Unknown/unknown	BTV-XJ1407/BTV-XJ1407	KR815991.1/AMM44552.1	93.63/97.24	100/100
2/VP2 (LR877338)	BTV-26/BTV-26	KUW2010-02/KUW2010-02	HM590642.1/AED99447.1	68.13/67.01	100/100
3/VP3 (LR877339)	Unknown/unknown	V196-XJ-2014/V196-XJ-2014	KX695172.1/ASW41948.1	92.65/98.45	100/100
4/VP4 (LR877340)	Unknown/unknown	BTV-XJ1407/BTV-XJ1407	KR085414.1/AMM44546.1	90.49/97.83	100/100
5/NS1 (LR877341)	Unknown/unknown	BTV-XJ1407/BTV-XJ1407	KR085415.1/AMM44547.1	81.97/88.20	99/99
6/VP5 (LR877342)	Unknown/unknown	SPvvvv-02/SPvvvv-02	MN723875.1/QGW56800.1	76.15/86.50	100/100
7/VP7 (LR877343)	Unknown/unknown	V196-XJ-2014/V196-XJ-2014	KX695176.1/ASW41952.1	91.62/99.71	100/100
8/NS2 (LR877344)	Unknown/unknown	V196-XJ-2014/BTV-XJ1407	KX695177.1/AMM44549.1	87.01/92.35	100/100
9/VP6 (LR877345)	Unknown/unknown	BTV-XJ1407/BTV-XJ1407	KR085418.1/AMM44550.1	88.38/87.23	100/100
10/NS3 (LR877346)	Unknown/unknown	V196-XJ-2014/V196-XJ-2014	KX695179.1/ASW41956.1	95.07/98.25	100/100
1/VP1 (LR877347)	Unknown/unknown	BTV-XJ1407/BTV-XJ1407	KR815991.1/AMM44552.1	93.22/97.16	100/100
2/VP2 (LR877348)	Unknown/unknown	BTV-XJ1407/BTV-XJ1407	KR061882.1/AMM44543.1	84.95/87.76	100/100
3/VP3 (LR877349)	Unknown/unknown	V196-XJ-2014/V196-XJ-2014	KX695172.1/ASW41948.1	93.20/98.89	100/100
4/VP4 (LR877350)	Unknown/unknown	BTV-XJ1407/BTV-XJ1407	KR085414.1/AMM44546.1	90.59/97.98	100/100
5/NS1 (LR877351)	Unknown/unknown	BTV-XJ1407/BTV-XJ1407	KR085415.1/AMM44547.1	89.75/93.65	100/99
6/VP5 (LR877352)	Unknown/unknown	BTV-XJ1407/BTV-XJ1407	KR061883.1/AMM44544.1	86.72/94.87	100/100
7/VP7 (LR877353)	Unknown/unknown	BTV-XJ1407/V196-XJ-2014	KR085416.1/ASW41952.1	92.86/100.00	100/100
8/NS2 (LR877354)	Unknown/unknown	BTV-XJ1407/BTV-XJ1407	KR085417.1/AMM44549.1	87.57/94.05	100/100
9/VP6 (LR877355)	Unknown/unknown	BTV-XJ1407/BTV-XJ1407	KR085418.1/AMM44550.1	89.29/87.23	100/100
10/NS3 (LR877356)	BTV-27/unknown	BTV-27-FRA2014-v02/BTV-X ITL2015	KU760996.1/APC23697.2	83.33/93.45	100/100
1/VP1 (LR877358)	Unknown/unknown	BTV-XJ1407/BTV-XJ1407	KR815991.1/AMM44552.1	93.07/97.24	100/100
2/VP2 (LR877359)	Unknown/unknown	BTV-XJ1407/BTV-XJ1407	KR061882.1/AMM44543.1	72.80/75.21	100/100
3/VP3 (LR877360)	Unknown/unknown	V196-XJ-2014/V196-XJ-2014	KX695172.1/ASW41948.1	91.13/98.78	100/100
4/VP4 (LR877361)	Unknown/unknown	BTV-XJ1407/BTV-XJ1407	KR085414.1/AMM44546.1	92.66/98.45	100/100
5/NS1 (LR877362)	Unknown/unknown	BTV-XJ1407/BTV-XJ1407	KR085415.1/AMM44547.1	90.87/94.74	99/99
6/VP5 (LR877363)	BTV-27/unknown	BTV-27-FRA2014-v02/BTV-X ITL2015	KU760992.1/APC23693.2	80.74/89.67	99/99
7/VP7 (LR877364)	Unknown/unknown	V196-XJ-2014/V196-XJ-2014	KX695176.1/ASW41952.1	92.67/100.00	100/100
8/NS2 (LR877365)	Unknown/unknown	V196-XJ-2014/BTV-XJ1407	KX695177.1/AMM44549.1	87.66/94.62	100/100
9/VP6 (LR877366)	Unknown/unknown	BTV-XJ1407/BTV-XJ1407	KR085418.1/AMM44550.1	88.89/87.54	100/100
10/NS3 (LR877367)	Unknown/unknown	V196-XJ-2014/V196-XJ-2014	KX695179.1/ASW41956.1	94.64/98.69	100/100

* Query cover of >85.

**Table 5 viruses-13-00042-t005:** Overview of atypical BTV strains belonging to different ‘putative novel serotypes’ according to their segment 2 genome sequence. Nearest neighbours with query cover >85% were listed, and the accession numbers are in brackets. (Putative novel) serotypes were sorted chronologically to their collection date of the first identified BTV strain of the respective serotype.

Putative Novel Serotypes	Serotype	Country of Origin	BTV Strain Designation	Successful Virus Isolation	Accession No.— Seg2 nt	Nearest Neighbour/Homology in Seg-2 (% Query Cover)	Nearest Neighbour of Different Serotype/Homology in Seg-2 (% Query Cover)	Date of Sequence Submission	Collection Date	Publication
25	BTV-25	Switzerland	BTV-TOV	no	EU839840	BTV-GER2018 (LR798442)/83.48% (100)	BTV-27-v03 (KU760998)/74.09% (100)	20-JUN-2008	14-DEC-2007	[11]
25	unknown	Italy	BTV-Z ITL2017	no	MF673721	BTV-Z ITL2017 (MF673721)/92.53% (100)	BTV-27-v02 (KU760988.1)/74.27% (99)	16-AUG-2017	04-JUL-2017	[41]
25	unknown	Germany	BTV-25GER2018	yes	LR798442	BTV-25GER2018 (LR798442)/92.53% (100)	BTV-27-v02 (KU760988.1)/74.54% (98)	18-MAY-2020	JUL-2018	[33]
26	BTV-26	Kuwait	BTV-KUW2010/02	yes	HM590642	SPvvvv/02 (MN723871.1)/ 72.45% (99)	SPvvvv/02 (MN723871.1)/72.45% (99)	25-JUN-2010	FEB-2010	[7]
27	BTV-27	France	BTV-27v01	yes	KM200718	BTV-27v03 (KU760998)/92.63% (100)	BTV-X-ITL2015 (KY365755.1)/75.38% (100)	15-JUL-2014	16-JAN-2014	[8]
27	BTV-27	France	BTV-27v02	yes	KU760988	BTV-27v03 (KU760998)/92.80% (100)	BTV-X-ITL2015 (KY365755.1)/75.50% (100)	24-FEB-2016	2014	[42]
27	BTV-27	France	BTV-27v03	yes	KU760998	BTV-27v02 (KU760988)/92.80% (100)	BTV-X-ITL2015 strain 33531(KY365755.1)/75.29% (100)	24-FEB-2016	2014	[42]
28	unknown	Israel	BTV-28/1537/14	yes	MH559807	SPvvvv/03 (MN723881.1)/99.86% (100)	BTV-11 (JN003580.1)/64.47% (87)	02-JUL-2018	2014	[15,16]
28	unknown	Israel	SPvvvv/03	yes	MN723881.1	BTV-28/1537/14 (MH559807)/99.86% (100)	BTV-11 (JQ972862.1)/64.48% (88)	21-NOV-2019	2014	[17]
29	Unknown	Israel	SPvvvv/02	yes	MN723871	BTV-26 (HM590642.1)/72.45% (100)	BTV-26 (HM590642.1)/72.45% (100)	21-NOV-2019	2014	[17]
30	unknown	China	BTV-XJ1407	yes	KR061882	BTV-MNG2/2016 (LR877348)/85.03% (100)	BTV-X-ITL2015 (KY365755.1)/76.40% (100)	04 APR 2015	14 JUL 2014	[13]
30	unknown	Mongolia	BTV-MNG2/2016	yes	LR877348	BTV-XJ1407 (KR061882)/85.03% (100)	TOV (EU839840)/74.73% (98)	12 OCT 2020	2016	This study
31	unknown	China	V196/XJ/2014	yes	KX695171	BTV-25 GER2018 (LR798442.1)/66.36% (85)	BTV-25 GER2018 (LR798442.1)/66.36% (85)	12-AUG-2016	20 SEP 2014	Direct submission
32	unknown	Italy	BTV-X-ITL2015	no	KX234079	BTV-X ITL2015 strain 33,531 (KY365755.1)/99.83% (100)	BTV-XJ1407 (KR061882.1)/76.35% (100)	16-DEC-2016	30-OCT-2015	[14]
32	unknown	Italy	BTV-X ITL2015 strain 33531	no	KY365755.1	BTV-X-ITL2015 (KX234079)/99.83% (100)	BTV-XJ1407 (KR061882.1)/76.40% (100)	17-DEC-2016	30-OCT-2015	[14]
33	unknown	Mongolia	BTV-MNG3/2016	yes	LR877368	BTV-XJ1407 (KR061882.1)/72.42% (100)	BTV-XJ1407 (KR061882.1)/72.42% (100)	12 OCT 2020	2016	This study
34	unknown	Tunisia	BTV-Y TUN2017	no	MF124283	BTV-24 (MN710201.1)/65.76% (85)	BTV-24 (MN710201.1)/65.76% (85)	17-MAY-2017	03-JAN-2017	[19]
35	unknown	Mongolia	BTV-MNG1/2018	yes	LR877348	BTV-26 (HM590642)/68.13% (100)	BTV-26 (HM590642)/68.13% (100)	12 OCT 2020	2018	This study

## Data Availability

The data presented in this study are available on request from the corresponding author.

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
