# Peer review of "Putative Novel Serotypes ‘33’ and ‘35’ in Clinically Healthy Small Ruminants in Mongolia Expand the Group of Atypical BTV"

_viruses, 2020, doi:10.3390/v13010042_

Round 1

Reviewer 1 Report

The manuscript entitled “Novel BTV genotypes ‘33’ and ‘35’ in clinically healthy small ruminants in Mongolia expand the group of atypical BTV” reports the isolation of novel strains of BTV from Mongolia and the characterisation by molecular, in vitro and in vivo means.

The work is interesting and the findings are important to BTV surveillance in East Asia. The reviewer’s opinion is that the ‘sampling in Mongolia’ section should be slightly expanded to include a map to indicate location samples were collected from, number and percent of animals positive on RT-qPCR, where the co-infection was detected. The type of livestock which surveillance was undertaken, be domesticated or nomadic, should be indicated. Complimentary to the virological investigation, serological findings should be included, or an explanation provided as to why not present in this report.

The cell culture section should precede animal experiment in order to provide greater flow and narrative for the rationale of conducting in vivo trial for MNG2/2016. This section could be difficult to read as there is many types of cells used for isolation. This section would be benefit with a table to allow readers to understand the process involved. It is unclear if all 3 samples were subjected to a first pass in KC cells. The authors should provide a brief explanation on why embryonated eggs were not used for virus isolation, despite it is one of the isolation method listed on the OIE manual.

Although there is extensive genomic analysis as provided in Figure 1 and Table 1, the report fails to provide description of those results in the text, which is important especially for keen readers to understand the interpretation of the findings by the authors. In particular, the reviewer found the divergence of MNG2 and MNG1-MNG3 at Seg-10 very interesting and should be discussed in regards to the potential incursion, re-assortment, or evolution with vector species. On the other hand, the genomic analysis section has significant overlap of contents with the genotype classification section. The results should be presented in a more logical format, which the reviewer proposed the following order: comparison of nearest neighbours, followed by comparison of MNG1 to 3 (including each segments), analysis of genotype as an alternative to serotype, and finally nucleotype for evolutionary lineages.

Apart from the points mentioned above, this manuscript should be reviewed for editorial issues. The reviewer is particularly concern with the lack of conventional description and order in the materials and methods section. The editorial comments suggested by the reviewer is non-exhaustive and it is the authors’ responsibility to provide quality write up for the journal.

The reviewer recommends that a major revision for this manuscript prior to a further consideration.

Author Response

Reviewer 1:Comments and Suggestions for Authors
The manuscript entitled “Novel BTV genotypes ‘33’ and ‘35’ in clinically healthy small ruminants in Mongolia expand the group of atypical BTV” reports the isolation of novel strains of BTV from Mongolia and the characterisation by molecular, in vitro and in vivo means.The work is interesting and the findings are important to BTV surveillance in East Asia. The reviewer’s opinion is that the ‘sampling in Mongolia’ section should be slightly expanded to include a map to indicate location samples were collected from, number and percent of animals positive on RT-qPCR, where the co-infection was detected. The type of live stock which surveillance was undertaken, be domesticated or nomadic, should be indicated.The type of livestock is now indicated as nomadically. Furthermore,we added a table showing the number and percent of animals positive in multiple RT-qPCR assays, indicating co-infections. The map of Mongolia with the sampling location sides is added as well.Complimentary to the virological investigation, serological findings should be included, or an explanation provided as to why not present in this report.The serological findings of the study are added to the Supplementary material and are indicated in the result chapter.Only for the second shipment serum was available for investigation in the cELISA. The aim of the presented study was the identification and characterisation of BTV strain from Mongolia. The more value of the serological data to our study goal is very little, as the study designis not representative for serological surveillance or epidemiological conclusions. A direct connection of the positive cELISA results to the infection with the Mongolian strains is furthermore unclear and the serum samples were in not in good quality and volume, which excludes further analysis in the VNT. This explanation is given in the supplement part as well.The cell culture section should precede animal experiment in order to provide greater flow and narrative for the rationale of conducting in vivo trial for MNG2/2016.We thank the reviewer asking for more explanation about the purpose of the in vivo trial with MNG2/2016. We provided now additional explanation in the Material& Methods part, on the beginning of the Experimental infection section and explained the purpose of the animal trial with the following sentence:The experimental inoculation with BTV-MNG2/2016 was conducted for receiving fresh EDTA blood for the isolation ofBTV-MNG2/2016 in cell culture.”Changing the order of the sections would diminish the reading flow in the result part and from our perspective the order of sampling in Mongolia’, ‘Animal experiments’, ‘The Mongolian strain in cell culture’, followed by ‘Virus neutralization’ should not be changed.However, we implemented the suggestion of the reviewer to make the purpose of the animal trail from the beginning of the manuscript clearer.We hope its now more rationale.This section could be difficult to read as there is many types of cells used for isolation. This section would be benefit with a table to allow readers to understand the process involved. It is unclear if all 3 samples were subjected to a first pass in KC cells.The reviewer got confused with the procedure of the virus isolation, specifically with the used cell lines. The reviewer suspected a first pass in KC cells, which was not performed for any of the three strains.We added in the first sentence the cell line BSR, which was used for virus isolation.The virus isolation procedure is now followed by the conformation of the three serial passages via RT-qPCR and by the conformation via immunofluorescence. Thus, this immunofluorescence part we now forwarded before the virus propagation part. Furthermore,we separated the virus propagation part (which includes the KC cell line)with a line break from the virus isolation part. Furthermore, we added more details to the virus isolation part and repeated that virus isolation was done on BSR cells. We furthe rmore added graphic about the virus isolation procedure.The authors should provide a brief explanation on why embryonated eggs were not used for virus isolation, despite it is one of the isolation method listed on the OIE manual.Previous studies dealing with virus isolation of atypical BTV strains did not show success on ECE like for BTV-TOV or BTV-Y TUN2017 [1, 2]. However, the BTV-28 strains, which has high similarities with the classical BTV strains could be isolated via ECE [3], like for BTV-26 the virus isolation via ECE was conducted successfully [4]. However, in our laboratory we were for the first time able to isolate a BTV-25 related strain on BSR cells by using ‘fresh’ EDTA blood with the same method as described for BTV-MNG2/2016 [5]. Although there is extensive genomic analysis as provided in Figure 1 and Table 1, the report fails to provide description of those results in the text, which is important especially for keen readers to understand the interpretation of the findings by the authors. In particular, the reviewer found the divergence of MNG2 and MNG1-MNG3 at Seg-10 very interesting and should be discussed in regards to the potential incursion, re-assortment, or evolution with vector species. We added descriptions about the phylogenetic trees and BLAST table for all segments and in particular the requested segment-10. In result part added information about the phylogenetic trees and table 4: ‘The phylogenetic tree of segment 2 shows BTV-MNG2/2016 in close proximity to the BTV-XJ1407 strain and BTV-MNG3/2018 more distant but clearly in between the other atypical BTV strains. However, BTV-MNG1/2016 is in the same cluster as BTV-26 and SPvvvv/02. The nearest neighbour in the BLAST search are in line with the findings of the phylogenetic trees, showing the nearest neighbour of BTV-MNG2/2018 and BTV-MNG3/2016 to be BTV-XJ1407, whereas for BTV-MNG1/2016 the nearest neighbour was found to be BTV-26 in both nt and aa. For segment 6, the BTV-MNG2/2016 strains clustered with BTV-XJ1407, which is line with the BLAST result in both nt and aa. The BTV-MNG1/2018 and BTV-MNG3/2016 strains are more distant to other strains and gathering with BTV-26, SPvvvv/02, BTV-Y TUN2017, BTV-X ITL2015 and two variants of BTV-27 in the phylogenetic trees. The BLAST reflects the phylogenetic tree and revealed the nearest neighbour in nt and aa for BTV-MNG1/2018 to be SPvvvv/02, whereas for BTV-MNG3/2016 the nearest neighbour in nt was BTV-27v02 and in aa BTV-X ITL2015. The phylogenetic trees for segment 1, 3, 4, 5, 7, 8 and 9 revealed, that all three Mongolian strains are close to each other and to the two Chinese strains BTV-XJ1407 and V/196/XJ/2014, which is in line with the BLAST research of finding BTV-XJ1407 and V/196/XJ/2014 as nearest neighbours in nt and aa.In segment 10, BTV-MNG3/2016 and BTV-MNG1/2018 showed proximity with each other and with V/196/XJ/2014, whereas BTV-MNG2/2016 is close to BTV-27 v02 and BTV-X ITL2015.Likewise, the nearest neighbour found in BLAST research for BTV-MNG1/2018 and BTV-MNG3/2016 was V/196/XJ/2014 in nt and aa, whereas for BTV-MNG2/2016 in nt BTV-27v02 and in aa BTV-X ITL2015.’In discussion part:The phylogenetic analysis revealed that the segments 1, 3, 4, 5, 7, 8 and 9 of all three Mongolian strains are close related to BTV-XJ1407 and V/196/XJ/2014. This is not surprising as BTV-XJ1407 and V/196/XJ/2014 are both detected in China, as neighbouring country of Mongolia [6]. However, in the segments 2,6 and 10 some of the three Mongolian strains did reveal the nearest neighbour’s amongst BTV strains detected in more distant countries like Israel (SPvvvv/02 [7]), Kuwait (BTV-26 [8]), France (BTV-27v02 [9]) and Italy (BTV-X ITL2015). The global spread of BTV is a complex, multifactorial development driven by environmental and anthropogenic factors [10]. Furthermore, reassortment events can happen and are not limited to phylogenetically related viruses [11]. This might have occurred for the mentioned segments 2, 6 and 10 of the mentioned Mongolian strains.`On the other hand, the genomic analysis section has significant overlap of contents with the genotype classification section. The results should be presented in a more logical format, which the reviewer proposed the following order: comparison of nearest neighbours, followed by comparison of MNG1 to 3 (including each segments), analysis of genotype as an alternative to serotype, and finally nucleotype for evolutionary lineages.We adapted the structure of the genome analysis part as the reviewer suggested starting with the comparison of the nearest neighbour, followed by the detailed description of all segments of the three Mongolian strains and finally with the serotype classificationpart. Hence, as now the phylogenetically analysis part of all ten segmentsis clearlyextended, we refrained the evolutionary lineages and nucleotype part, which is just referringto segment 6 and would lead to an overlap of contents as critickedbythe reviewer.

Apart from the points mentioned above, this manuscript should be reviewed for editorial issues. The reviewer is particularly concern with the lack of conventional description and order in the materials and methods section. The editorial comments suggested by the reviewer is non-exhaustive and it is the authors’ responsibility to provide quality write up for the journal.The reviewer recommends that a major revision for this manuscript prior to a further consideration.We hope the revised version of the manuscript will close the gaps in the description in the used material and method and improvesthe presentation of the generated results.1.Lorusso, A., et al., Analysis of bluetongue serotype 3 spread in Tunisia and discovery of a novel strain related to the bluetongue virus isolated from a commercial sheep pox vaccine.Infect Genet Evol, 2018. 59: p. 63-71.2.Planzer, J., et al., In vivo and in vitro propagation and transmission of Toggenburg orbivirus.Res Vet Sci, 2011. 91(3): p. e163-8.3.Bumbarov, V., et al., Detection and isolation ofBluetongue virus from commercial vaccine batches.Vaccine, 2016. 34(28): p. 3317-23.4.Batten, C.A., et al., Bluetongue virus serotype 26: infection kinetics and pathogenesis in Dorset Poll sheep.Vet Microbiol, 2012. 157(1-2): p. 119-24.5.Ries, C., etal., Isolation and Cultivation of a New Isolate of BTV-25 and Presumptive Evidence for a Potential Persistent Infection in Healthy Goats.Viruses, 2020. 12(9).6.Sun, E.C., et al., Emergence of a Novel Bluetongue Virus Serotype, China 2014.Transbound Emerg Dis, 2016. 63(6): p. 585-589.7.Rajko-Nenow, P., et al., Complete Coding Sequence of a Novel Bluetongue Virus Isolated from a Commercial Sheeppox Vaccine.Microbiol Resour Announc, 2020. 9(10).8.Maan, S., et al., Novel bluetongue virus serotype fromKuwait.Emerg Infect Dis, 2011. 17(5): p. 886-9.9.Schulz, C., et al., Bluetongue virus serotype 27: detection and characterization of two novel variants in Corsica, France.J Gen Virol, 2016. 97(9): p. 2073-83.10.Samy, A.M. and A.T. Peterson, Climate Change Influences on the Global Potential Distribution of Bluetongue Virus.Plos One, 2016. 11(3).11.Maan, N.S., et al., The Genome Sequence of Bluetongue Virus Type 2 from India: Evidence for Reassortment between Eastern and Western Topotype Field Strains.Journal of Virology, 2012. 86(10): p. 5967-5968

Reviewer 2 Report

The manuscript describes the isolation of putative novel serotypes of Bluetongue virus (BTV) from small ruminants in Mongolia. Although animal husbandries flourish in the country, information of arbovirus infections in ruminants has been quite limited so far. The manuscript partially fills the knowledge gaps. Identification of novel BTV serotypes in small ruminants expands our knowledge on the molecular biology, pathogenicity and epidemiology of BTV. Table 4 in the manuscript will enhance our understanding for classification of atypical BTVs. However, “genotype” of BTV might not be acceptable for many readers. This term will create some confusion on BTV taxonomy. As the authors stated, the new BTV strains fulfill the criteria as a novel serotype. Therefore, the strains could be defined as putative novel serotypes of BTV. The authors’ proposal would require the approval of other scientist who study on BTV/other orbiviruses. Please address the following detail comments.

Title: ‘Putative novel serotypes’ might be better.

Line 20-24: These sentences seems to be abrupt. ‘34’ and ‘31’ should be changed to ‘35’ and ‘30’, respectively.

Line146-150: More detailed description would be necessary.

Line 156: ‘2’ should be displayed as subscript. ‘CPE’ first appeared here.

Line 174: ‘2’ should be displayed as superscript.

Line 176: Tissue culture infectious dose (TCID50)? Please show the method for virus titration.

Line 345: UPGMA is appeared to be generally unused for phylogenetic analysis at present. The maximum likelihood with best fit model would be better for the analyses, if there is no good reason.

Line 257-258: More detailed description for the BTV-positive samples should be necessary. Did the authors check the samples by PCR assays which can discriminate each classical BTV serotypes? Did the authors conduct VNT with classical BTV serotypes for serum samples?

Line 268-269: Could the authors show VNT titer(s) in the experimentally infected goat?

Line 283; The replication of BTV-MNG2/2016 and BTV-3MNG/2018 in the Culicoides cell line should be checked. The finding will support that they are ‘atypical BTVs’.

Line 348-364: The paragraph is very important for future classification of BTV. However, serotyping is still general at present. ‘Genotyping’ might be treated carefully in the manuscript.

Author Response

Reviewer 2

Comments and Suggestions for Authors

The manuscript describes the isolation of putative novel serotypes of Bluetongue virus (BTV) from small ruminants in Mongolia. Although animal husbandries flourish in the country, information of arbovirus infections in ruminants has been quite limited so far. The manuscript partially fills the knowledge gaps. Identification of novel BTV serotypes in small ruminants expands our knowledge on the molecular biology, pathogenicity and epidemiology of BTV. Table 4 in the manuscript will enhance our understanding for classification of atypical BTVs. However, “genotype” of BTV might not be acceptable for many readers. This term will create some confusion on BTV taxonomy. As the authors stated, the new BTV strains fulfil the criteria as a novel serotype. Therefore, the strains could be defined as putative novel serotypes of BTV. The authors’ proposal would require the approval of other scientist who study on BTV/other orbiviruses. Please address the following detail comments. Title: ‘Putative novel serotypes’ might be better.èWe understand the reviewers concern and according to the reviewers suggestion, we changed in the whole manuscript the term ‘genotype’ to ‘putative novel serotype’.

Line 20-24: These sentences seems to be abrupt. ‘34’ and ‘31’ should be changed to‘35’ and ‘30’, respectively.èThe suggested improvements were implemented and the sentences were little changed to not being too abrupt.

Line 146-150: More detailed descriptionwould be necessary.èMore details are included for the description of the virus isolation part, as well as a graphic.Line 156: ‘2’ should be displayed as subscript. ‘CPE’ first appeared here.èThe ́2 ́is subscripted and the abbreviation CpE is explained.

Line 174: ‘2’ should be displayed as superscript.èThe 2 of T1700 cm2 is now superscripted.

Line 176: Tissue culture infectious dose (TCID50)? Please show the method for virus titration.èWe replaced in the whole manuscript TCID50with CCID50as the cell culture infectious dose was calculated. TCID50 was an error and not calculated in the manuscript.

Line 345: UPGMA is appeared to be generally unused for phylogenetic analysis at present. The maximum likelihood with best fit model would be better for the analyses, if there is no good reason.

èAll 10 phylogenetic trees were remade with the maximum likelihood tree built method like suggested.

Line 257-258: More detailed description for the BTV-positive samples should be necessary. Did the authors check the samples by PCR assays which can discriminate each classical BTV serotypes?Did the authors conduct VNT with classical BTV serotypes for serum samples?èWe discriminated between classical and atypical BTVs by analysing the partial sequences of the Pan-BTV-S10 postive samples. The qualitiy of the serum samples but also the non availablity of several serum samples, was not allowing the performance of VNT tests with all the Pan-BTV-S10 positive BTV samples. However, more details to the positive samples are given: ‘The partial VP2 sequences of the Pan-BTV-S10 RT-qPCR positive samples, revealed that no classical BTV serotype could be found, but three different novel BTV-strains, so called BTV-MNG1/2018, BTV-MNG2/2016, and BTV-MNG3/2016.’

Line 268-269: Could the authors show VNT titer(s) in the experimentally infected goat?èAs suggested by the reviewer, we additionally performed the VNT with the serum of the experimentally infected goat with a negative result.

Line 283; The replication of BTV-MNG2/2016 and BTV-3MNG/2018 in the Culicoides cell line should be checked. The finding will support that they are ‘atypical BTVs’.èIndeed, this comment of the reviewer is of very big interest for us as well. Currently, we plan an extensive investigation of virus propagation on a wide spectrum of mammalian and insect derived cells, not only KC cells. Furthermore, we will include not only the remaining two Mongolian strains, but a whole group of atypical strains available in our laboratory for supporting the difference of atypcial BTV strains towards classical BTV strains. However, these results would completely overload the here presented manuscript. Nevertheless, the molecular analysis left no doubt, that all three Mongolian strains are part of the atypical BTVs.

Line 348-364: The paragraph is very important for future classification of BTV. However, serotyping is still general at present. ‘Genotyping’ might be treated carefully in the manuscript.èThe term genotype is removed from the manuscript. The alternative ‘putative novel atypical serotype’ suggested by the reviewer is used instead.

Reviewer 3 Report

Review report manuscript no. viruses-987698

Title: “Novel BTV genotypes ‘33’ and ‘35’ in clinically healthy small ruminants in Mongolia expand the group of atypical BTV”.

Authors: Christina Ries, Tumenjargal Sharav, Erdene-Ochir Tseren-Ochir, Martin Beer, Bernd Hoffmann.

This authors describes the detection and investigation of new BTV variants in healthy small ruminants in Mongolia. These new BT viruses were extensively studied by cell culturing, rising rabbit antisera, and experimental infection of goats. Conventional serotyping by neutralization assays was not possible by the “two-way” method, since antisera raised in rabbits for these new BT viruses contained low nAB titres. Still, antisera for BTV1-24 were not able to neutralize these viruses, suggesting that these are new serotypes or serotypes belonging to existing new serotypes showing the same phenomenon.

Coding regions of these new variants were completely sequenced and compared to representatives of serotypes and nucleotypes. Phylogenetic analysis was particularly focused on genome segment 2 coding for serotype immunodominant VP2 protein as genetic homology is strongly associated to serotypes as shown for classical BTV1-24. The authors claim two variants as novel “genotypes”. Further, the novel variants were “classified” as atypical BT viruses based on their in vitro growth characteristics, virus isolation in small ruminants, and the high homology with atypical BTV-25 to 27.

Major comments:

  • The manuscript is well written, however, this reviewer is not convinced regarding the claim of two novel genotypes. Consequently, the impact of the study is too low for publication in “Viruses”. The research is very well performed but the interpretation of results is not correct. The title must be significantly weakened like “Expansion of the group of atypical bluetongue viruses by variants in clinically healthy small ruminants in Mongolia”.
  • If I understand correctly, BTV-MNG1/2018 is proposed as new genotype 35 (and maybe a new serotype) based on 0.3% nucleotide difference (68.1 treshold 68.4) in Seg-2. (Table 3). This means a difference of appr. 9 nucleotides in the VP2-ORF of about 3000 nucleotides!!! To my opinion, this difference is too small to claim a novel genotype. Even more, if more isolates will be sequenced, all variants between two genotypes could be found. So, these results will feed the discussion to group and to subgroup virus variants.
  • Further in L360-364: “According to this high similarity on genome basis, BTV-MNG2/2016 is part of the same genotype 30, like the Chinese BTV-XJ1407. The nucleotide sequence of the segment-2 of BTV-MNG3/2016 matches to 72.4% with BTV-XJ1407. The closest neighbour on amino acid level is likewise BTV-XJ1407 with 75.2%. Subsequently, BTV-MNG3/2016 …”??? To my opinion this homology is high enough to also group this isolate to genotype 30!! However, Figure 1 (VP2) is not in agreement to this, and suggests that this variants is located between 25 and 26 and could be indeed a novel genotype??

Minor comments:

Abstract

L23-24: “The BTV-MNG1/2018 isolate forms ‘genotype’ 33, the BTV-MNG3/2016 ‘genotype’ 34, whereas the BTV-MNG2/2016 strain belongs to the same genotype ‘31’ as BTV-XJ1407 from China.” Please check isolate numbers and proposed genotype numbers. These do not correspond to the last paragraph of the introduction section and Table 4.

Intro

For the readability, this reviewer recommend to forward the reader to Table 4 in the paragraphs describing the previously proposed novel serotypes/genotypes.

L44:add the reference number of Wright.

L46-55: This part can be written in a more understandable manner.

L54-55: for segment 2/VP2 or the entire virus?? please describe.

Materials and methods

CCID50/ml and TCID50/ml were both used. Is there is a difference?

L100-105: Apparently, shipped whole blood samples were not frozen as red blood cells could be washed to remove neutralizing Abs. Please add shipment conditions.

L120: Why inactivation of BTV prior inoculation of rabbits?

L137-163: add official references of cell lines (like ATCC number) and ingredients (supplier) for independent reproducibility.

L168: the cut-off given by the supplier is 45%. It should be argued when another cut-off value was used? Consequently, according to the suppliers’ threshold only 31 dpi of goat #18 was positive (Table 2).

L190: Where are results with this PCR test?

L205: remove ‘degree’

L210: why not BTV-25 tested too? Apparently there is a German isolate (BTV-25GER-2018)?

Table 1: primer and probe names suggest the serotype number but these do not correspond the isolate number as mentioned in the text and Table 4. This is confusing.

Table 2. Cq values but in the text Ct value was used. Which RT-PCR test was used??

L281: please change a CpE to CpE.

L292-293: please correct like …  were unable to neutralize all…. (as sera cannot be neutralized by virus).

Author Response

Reviewer 3

Comments and Suggestions for Authors

This authors describes the detection and investigation of new BTV variants in healthy small ruminants in Mongolia. These new BT viruses were extensively studied by cell culturing, rising rabbit antisera, and experimental infection of goats. Conventional serotyping by neutralization assays was not possible by the “two-way” method, since antisera raised in rabbits for these new BT viruses contained low nAB titres. Still, antisera for BTV1-24 were not able to neutralize these viruses, suggesting that these are new serotypes or serotypes belonging to existing new serotypes showing the same phenomenon.

Coding regions of these new variants were completely sequenced and compared to representatives of serotypes and nucleotypes. Phylogenetic analysis was particularly focused on genome segment 2 coding for serotype immunodominant VP2 protein as genetic homology is strongly associated to serotypes as shown for classical BTV1-24. The authors claim two variants as novel “genotypes”. Further, the novel variants were “classified” as atypical BT viruses based on their in vitro growth characteristics, virus isolation in small ruminants, and the high homology with atypical BTV-25 to 27.

Major comments: The manuscript is well written, however, this reviewer is not convinced regarding the claim of two novel genotypes. Consequently, the impact of the study is too low for publication in “Viruses”. The research is very well performed but the interpretation of results is not correct. The title must be significantly weakened like “Expansion of the group of atypical bluetongue viruses by variants in clinically healthy small ruminants in Mongolia”. If I understand correctly, BTV-MNG1/2018 is proposed as new genotype 35 (and maybe a new serotype) based on 0.3% nucleotide difference (68.1 treshold 68.4) in Seg-2. (Table 3). This means a difference of appr. 9 nucleotides in the VP2-ORF of about 3000 nucleotides!!! To my opinion, this difference is too small to claim a novel genotype. Even more, if more isolates will be sequenced, all variants between two genotypes could be found. So, these results will feed the discussion to group and to subgroup virus variants. Further in L360-364: “According to this high similarity on genome basis, BTV-MNG2/2016 is part of the same genotype 30, like the Chinese BTV-XJ1407. The nucleotide sequence of the segment-2 of BTV-MNG3/2016 matches to 72.4% with BTV-XJ1407. The closest neighbour on amino acid level is likewise BTV-XJ1407 with 75.2%. Subsequently, BTV-MNG3/2016 ...”??? To my opinion this homology is high enough to also group this isolate to genotype 30!! However, Figure 1 (VP2) is not in agreement to this, and suggests that this variants is located between 25 and26 and could be indeed a novel genotype??èThe reviewer did comment our research performance as very well, however the claim of two novel genotypes was not understood and therefore the reviewer criticised our conclusion up to questioning the whole negotiability of the data. The reviewer claims the three Mongolian strains as only variants. We thank the reviewer for this very important critic, as it shows us that we need to implement more and better explanation on why the two Mongolian strains are novel atypical

BTV serotypes and not variants. Therefore, we expanded the Table 5 with another column, which shows the very high inter-variant similarities of more than 80%. As the reviewer interpreted right, the BTV-MNG1/2018 and the BTV-MNG3/2016 sequences are on the border of the identity levels. However, they need to be declared as novel putative serotypes and not to variants as this low sequence identity is not comparable to the observed inter-variant similarity of more than 80 %. If the reviewer questions, that the two Mongolian strains are not novel serotypes, he also questions the declaration of BTV-27 as novel serotypes (and not to be BTV-25 variants). The reviewer further questions why BTV-X ITL2015 was not declared as variant of BTV-XJ1407 as well as SPvvvv/02 as variant of BTV-26.èWe expanded the classification of BTV-MNG3/2016 with the following explanation: ‘The nucleotide sequence of the segment-2 of BTV-MNG3/2016 matches to 72.8% with BTV-XJ1407. The closest neighbour on amino acid level is likewise BTV-XJ1407 with 75.2%. The nucleotide variation of 27.2 % between the nearest neighbour is close to the observed 28.5 % minimum variation in nt described by Maan, et al. The second nearest neighbour of BTV-MMNG3/2016 was found to be BTV-Z ITL2017 in nt with 72.4% identity and BTV-27v01 with 74.4% in aa. Hence, the second nearest neighbour BTV-27v01 is nearly as close as the nearest neighbour BTV-XJ1407, but they are of different serotypes. This is also reflected in the phylogenetic tree, where the BTV-MNG3/2016strain is on a distinct separate branch, not together with BTV-XJ1407 and BTV-MNG2/2016. Additionally, the sera of BTV-MNG3/2016 is not neutralizing BTV-MNG2/2016 (the same novel putative serotype as BTV-XJ1407) and the specific BTV-MNG3/2016 assays do not detect the other Mongolian strains. Therefore, we conclude, that BTV-MNG3/2016 represents a putative novel atypical serotype 33. A similar decision was made for BTV-27v01, which showed 73.0% identity in nt with BTV-25, which is bigger than the suggested 68.4% of Maan et al. [1]. BTV-27v01 was declared as novel serotype in agreement with the BTV community [2]. The calculations of Maan et al, [1]are good guidelines for classification of the novel atypical BTV strains, but are not to understand as exact values, as they are based on calculations of sequence identities at the respective time point of research. The inter-variant identities between strains are significantly higher than 80%, like for the three variants of BTV-25 and the three variants of BTV-27.’

Minor comments:Abstract

L23-24: “The BTV-MNG1/2018 isolate forms ‘genotype’ 33, the BTV-MNG3/2016 ‘genotype’ 34, whereas the BTV-MNG2/2016 strain belongs to the same genotype ‘31’ as BTV-XJ1407 from China.” Please check isolate numbers and proposed genotype numbers. These do not correspond to the last paragraph of the introduction section and Table 4.èThe numbers were changed

Round 2

Reviewer 1 Report

The authors have submitted first revision for the manuscript “Putative novel genotypes ‘33’ and ‘35’ in clinically healthy small ruminants in Mongolia expand the group of atypical BTV”.

The reviewer is dissatisfied with the re-submitted version that editorial comments provided on viruses-987698-peer-review-v1 comments.pdf on sticky notes have not been taken in account. Please kindly review that in line with your latest version of manuscript. It is attached under this general comment. 

Further epidemiological findings provided in the update in 2.1 and 3.1, albeit it is understood that this study is not tailored for such work, is really useful for demonstrating that atypical BTV is circulating in Mongolia at a low level. Has atypical BTV positive field-collected sheep blood subjected to virus isolation? If not, please state in text and provide rationale.

It is obvious that, particularly for the flow of the report, that single infection (supplementary Table 1) is more appropriate than co-infection (Table 2). It is also recommended that these two tables to be combined and to be described in main text.

In the animal experiment results section, this should be an area for describing transmission study for MNG2 strain. The inclusion of rabbit sera section for the assessment of BTV-specific antibodies for all 3 strains have made the section rather disjointed. That should fall within the same section with 3.2 virus neutralisation and be made under a broader title such as ‘serological profile’.

The authors have not addressed the question on why the results for animal study is provided first. It is understood that MNG2 could not be isolated by in vitro means from field-collected blood and hence an animal inoculation study was conducted. If the authors insisted that animal study has to be described beforehand of in vitro results, which would mean that in vivo study for MNG1 and MNG3, and under the presumption that they are three different strains of virus, has to be conducted as part of this report. Under this rationale, the justification for use of animals prior to the attempt of virus isolation has to be provided concisely.

There are also numerous errors in the manuscript that authors have to proof read diligently before submitting to the reviewer. The following are examples noted and are non-exhaustive:

  • The headings in results section are incorrect. There are two 3.1s and four 3.2s
  • Table 1 did not indicate which segment the primer probes are designed for
  • Figure 1 has only provided phylogenetic trees for 6 segments and yet figure description indicated 10 segments are provided.

The reviewer recommends that a major revision for any future consideration.

Author Response

Answer to comments of Reviewer 1

The authors have submitted first revision for the manuscript “Putative novel genotypes ‘33’ and ‘35’ in clinically healthy small ruminants in Mongolia expand the group of atypical BTV”.

The reviewer is dissatisfied with the re-submitted version that editorial comments provided on viruses-987698-peer-review-v1 comments.pdf on sticky notes have not been taken in account. Please kindly review that in line with your latest version of manuscript. It is attached under this general comment.

We kindly apologize for not considering the sticky notes. Unfortunately, we didn’t see the document in the first place. However, we are thankful for the comprehensive corrections and comments and integrated the sticky notes now into the manuscript.

In our re-submitted manuscript changes after the first revision are highlighted in yellow and changes after the second revision are highlighted in blue. According to the reviewer’s opinion we added more detailed information about the sampling areas and animals in Mongolia, changed the structure of the manuscript and re-placed genotype by putative novel atypical BTV. We believe, our manuscript profited from the reviewer comment substantially.Further epidemiological findings provided in the update in 2.1 and 3.1, albeit it is understood that this study is not tailored for such work, is really useful for demonstrating that atypical BTV is circulating in Mongolia at a low level. Has atypical BTV positive field-collected sheep blood subjected to virus isolation? If not, please state in text and provide rationale.

In general, blood samples for virus isolation were chosen according to the RT-qPCR results, quality and quantityof the sample. We didn’t select after goat or sheep origin.

In the first shipment, 14 blood samples were used for virus isolation. 12 goat EDTA blood samples and 2 sheep blood samples. Only one goat EDTA blood was successfullyin virus isolation (BTV-MNG3/2016)

In the second shipment, we used 27 EDTA blood samples for virus isolation. 26 samples were from goats, whereas one sheep bloodsamplewas used.However, we were only able to isolate the Mongolian strains in two goat bloods samples.(BTV-MNG1/2018 and BTV-MNG3/2016)

We added to the manuscriptin material and methods: `Fromthe first shipment 14 blood samples (of 12 goats and 2 sheep) were selected for virus isolation, whereas from the second shipment 27 blood samples (26 goats and 1 sheep) were chosen.

And in the result part: ‘In detail, virus isolation succeeded for BTV-MNG1/2018 from one field infected goat, for BTV-MNG2/216 from one experimental infected goat and for BTV-MNG3/2016 from two field infected goats. Hence the Mongolian strains were isolated in cell culture from goat EDTA blood samples, but not from sheep.It is obvious that, particularly for the flow of the report, that single infection (supplementary Table 1) is more appropriate than co-infection (Table 2). It is also recommended that these two tables to be combined and to be described in main text.

We changed the Supplementary table S1 into the main textas table 2.In the animal experiment results section, this should be an area for describing transmission study for MNG2 strain. The inclusion of rabbit sera section for the assessment of BTV-specific antibodies for all 3 strains have made the section rather disjointed. That should fall within the same section with 3.2 virus neutralisation and be made under a broader title such as ‘serological profile’.

We changed the structure in the M&M part as well as in the results and summarized in serological profile.The authors have not addressed the question on why the results for animal study is provided first. It is understood that MNG2 could not be isolated byin vitro means from field-collected blood and hence an animal inoculation study was conducted. If the authors insisted that animal study has to be described beforehand ofin vitro results, which would mean that in vivo study for MNG1 and MNG3, and under the presumption that they are three different strains of virus, has to be conducted as part of this report. Under this rationale, the justification for use of animals prior to the attempt of virus isolation has to be provided concisely.

We changed the order of the manuscript and in vitro results are now described before animal trail.There are also numerous errors in the manuscript that authors have to proof read diligently before submitting to the reviewer. The following are examples noted and are non-exhaustive:

The headings in results section are incorrect. There are two 3.1s and four 3.2s

Were changed

Table 1 did not indicate which segment the primer probes are designed for

Information is added

Figure 1 has only provided phylogenetic trees for 6 segments and yet figure description indicated 10 segments are provided.

Trees are addedThe reviewer recommends that a major revision for any future consideration.Comments to 12sticky notes (We applied in total 219comments or text corrections of reviewer 1)

1.Please provide a section on cells and viruses. The suggested sequence: cells & viruses, propagation in vitro, antisera production, animal trial, RT-PCR, NGS, VNT

Later in the manuscript the reviewer suggests to summarizeseveral chapters in serological profile, which we applied to the manuscript. To place VNT at the end would be contrary to the reviewer’sother comment. We inserted the chapter propagation in vitro

The new order is as followed: Sampling in Mongolia, Cell culture isolation of virus in vitro,Propagation in vitro,serological profile (Production of antisera in rabbits, ELISA, VNT),experimental infection of goats, RNA extraction and RT-qPCR, Sequence analysis

2.Line 19: does that imply MNG2 need to be adapted in mammalian host first?

All Mongolian strains originated from the mammalian host goat and were isolated on mammalian cell lines. The atypical BTV seem to be adapted very well to the mammalian host goat.

3.Line 74: This ref should be a conference proceeding from Kunming 1985

Unfortunately, we couldn’t find conference data from Kunming 1985

4.Line 102: It should be collected rather than deliveries. It can be collected from any years although delivered in 2018.

We added the collection years to shipment

2. We did work and analyzesamples separate in the shipments, hence we also presented them in the way we worked with them.Nevertheless, the collection years are indicated for both shipments, so no important information is missing.

5.Line 118: should provide another complimentary technique to confirm antibody production eg. WB

Proteins can be detected in the WB as well. However, there is no western blot validated and established for BTVor atypical BTVin our lab and therefore we didn’t use the technique.

6.Line 135: If there is residue of antibodies, wouldn't that affect efficiency of virus isolation in vitro?

The reviewer is correct. That’s why we used washed blood as well.

Reviewer 3 Report

No comments, The corrections and added information are sufficient

Author Response

We thank the reviewer for the gracious acceptance of our corrections.